# Transcriptome Analysis and Intraspecific Variation in Spanish Fir (*Abies pinsapo* Boiss.)

**DOI:** 10.3390/ijms23169351

**Published:** 2022-08-19

**Authors:** Francisco Ortigosa, Concepción Ávila, Lourdes Rubio, Lucía Álvarez-Garrido, José A. Carreira, Rafael A. Cañas, Francisco M. Cánovas

**Affiliations:** 1Grupo de Biología Molecular y Biotecnología, Departamento de Biología Molecular y Bioquímica, Universidad de Málaga, Campus Universitario de Teatinos, 29071 Málaga, Spain; 2Departamento de Botánica y Fisiología Vegetal, Universidad de Málaga, Campus Universitario de Teatinos, 29071 Málaga, Spain; 3Department of Ecology, Universidad de Jaén, Campus Las Lagunillas s/n., 23009 Jaén, Spain; 4Integrative Molecular Biology Lab, Universidad de Málaga, Campus Universitario de Teatinos, 29071 Málaga, Spain

**Keywords:** conifer, *Abies pinsapo*, transcriptome, intraspecific variation, chlorophyll fluorescence, nitrogen status

## Abstract

Spanish fir (*Abies pinsapo* Boiss.) is an endemic, endangered tree that has been scarcely investigated at the molecular level. In this work, the transcriptome of Spanish fir was assembled, providing a large catalog of expressed genes (22,769), within which a high proportion were full-length transcripts (12,545). This resource is valuable for functional genomics studies and genome annotation in this relict conifer species. Two intraspecific variations of *A. pinsapo* can be found within its largest population at the *Sierra de las Nieves* National Park: one with standard green needles and another with bluish-green needles. To elucidate the causes of both phenotypes, we studied different physiological and molecular markers and transcriptome profiles in the needles. “Green” trees showed higher electron transport efficiency and enhanced levels of chlorophyll, protein, and total nitrogen in the needles. In contrast, needles from “bluish” trees exhibited higher contents of carotenoids and cellulose. These results agreed with the differential transcriptomic profiles, suggesting an imbalance in the nitrogen status of “bluish” trees. Additionally, gene expression analyses suggested that these differences could be associated with different epigenomic profiles. Taken together, the reported data provide new transcriptome resources and a better understanding of the natural variation in this tree species, which can help improve guidelines for its conservation and the implementation of adaptive management strategies under climatic change.

## 1. Introduction

The Spanish fir (*Abies pinsapo* Boiss.) is an endemic conifer from southern Spain that belongs to the circum-Mediterranean firs group [1], which is cataloged as at risk of extinction on the International Union for the Conservation of Nature (UICN) Red List (as B1ab (i, ii, iii) + 2ab (i, ii, iii)) and in the Andalusian catalog of threatened species (as EN B1ab (iv) + 2ab (iv)). This species is considered isolated since the nearest populations of other fir species are found in northern Morocco (Rif range; *Abies maroccana*) and northern Spain (Pyrenees; *Abies alba*) [2]. This separation occurred prior to the Quaternary, according to palaeopalynological data [3]. The restricted distribution area (<3000 ha) of *Abies pinsapo* consists of scattered small patches in the western Baetic range that are distributed in three main areas: *Sierra de Grazalema* Natural Park (Cádiz), *Sierra de las Nieves* National Park (Malaga), and *Los Reales de Sierra Bermeja* Natural Reserve (Málaga). The national park contains 72.5% of the forest stands of Spanish fir that exist today [4]. This interesting conifer species lives in special ecological conditions under a particular subtype of Mediterranean climate that includes rainy seasons and severe warm and dry periods in summer.

In the last decade, considerable progress has been made in understanding the genetic and cellular processes that control the growth and survival of trees through the use of new genomic technologies that allow an overview of the set of genes and cellular processes involved in numerous aspects of development and adaptability to environmental stress factors [5,6,7]. The possibility of globally identifying, through genomic methodologies, genes functionally related to growth and the response to environmental stimuli in Spanish fir paves the way to implement restoration and conservation programs.

Intraspecific diversity can be defined as phenotypic variation between individuals of the same species and population [8]. This topic has been of great interest to scientists for decades, and plants exhibit significant intraspecific variation. Multiple lines of evidence strongly suggest that intraspecific variation has a positive effect on productivity as well as on resistance to biotic and abiotic stresses [9,10,11,12]. Intraspecific variation is a key component of community structure, ecosystem function, and adaptive evolution [8,13] and, therefore, in the resilience of the ecosystem. Intraspecific diversity can be the result of a variety of mechanisms, including local adaptation, artificial selection, parental conditions, and phenotypic plasticity [14]. In a genotypic framework, studies of intraspecific variation using several accessions of the model plant *Arabidopsis thaliana* revealed strategies for functional adaptation to climate change with consequences for important traits such as flowering, water-use efficiency, and total dry weight [15,16,17]. At the molecular level, genotypic variation can be the result of different mechanisms such as the existence of single nucleotide polymorphisms (SNPs) [18], the consequence of mutations in promotor sequences [19,20], the result of the effect of transposable elements (TEs) movement [21,22], or differential epigenomic mark deposition [12].

Although important discoveries have been made regarding intraspecific variation, most studies are focused on model plants [12,17,23,24,25,26]. Other plant species, particularly those that are endemic and have few tools and little information available for their study, have been scarcely studied. One suitable example of a poorly studied plant is the Spanish fir (*Abies pinsapo* Boiss.). In the National Park of *Sierra de las Nieves,* it is possible to find bluish and green needle phenotypes interspersed within the same area without the existence of spatial segregation between the variants (Figure 1). The aims of this work were to investigate, at the molecular and physiological levels, the possible causes of the above-mentioned difference, its biological significance, and the potential mechanisms behind the appearance of these two phenotypic variants.

## 2. Results

### 2.1. Spanish Fir Transcriptome Assembly Analysis

High-throughput cDNA sequencing was performed to define a reference transcriptome for Spanish fir (*A. pinsapo*). A combination of long reads obtained by pyrosequencing (454) and Illumina reads obtained from numerous samples were used for transcriptome assembly (Table 1). A preliminary analysis of the quality of the transcriptome assembly performed in this work was achieved using Full-LengtherNext (FLN) software. Table 2 shows the identification of 76,481 unigenes, of which only 8.29% were considered artifacts, resulting in 73,798 unigenes after resolving unmapped and misassembled transcripts and chimeras. Unigenes had an average length of 1799 base pairs (bp), and 36,503 were nonredundant transcripts based on ortholog identification (IDs). Remarkably, 20,135 full-length (FL) unigenes were reconstructed in the assembly, representing more than 50% of the total annotated unigenes (Table 2). In addition, the results derived from FLN analyses revealed that 37,021 unigenes did not possess significant homology to any other plant gene. These results may include new conifer genes but also artifactual assemblies. To distinguish between both possibilities, FLN includes a TestCode analysis [27] and a comparison with the noncoding RNA database (http://www.mirbase.org, accessed on 1 March 2021). Accordingly, it was estimated that at least 3409 nonredundant coding unigenes may be worth considering as putative new conifer genes. Finally, only 274 unigenes from this transcriptome assembly were determined to be candidate noncoding RNAs (ncRNAs). For these reasons, the minimum *A. pinsapo* transcriptome can be calculated as 19,086 unigenes with different IDs. In addition, when considering the 3409 unigenes without homology but with coding characteristics, plus the 274 noncoding RNAs, the *A. pinsapo* transcriptome could be composed of at least 22,769 unigenes (Table 2).

### 2.2. Differential Transcriptomics Revealed Different Gene Expression Patterns Depending on Tree Phenotype

To understand the molecular basis of the phenotypic differences between trees with bluish and green needles, hereafter bluish/green trees (Figure 1), transcriptome profiling was performed on the needles from bluish and green Spanish fir trees via RNA-seq. A total of 457 differentially expressed (DE) transcripts were identified after comparative transcriptome sequencing, and 173 and 284 DE transcripts were upregulated and downregulated, respectively, in the comparison between needles from bluish and green trees (bluish vs. green). All differential expression results are shown in Appendix A. Regarding the transcriptomics results, bluish trees exhibited enrichment in DE transcripts involved mainly in primary and secondary cell formation (Figure 2A), such as those encoding cellulose synthase (*ap_86459*, *ap_88208*), xyloglucan endotransglucosylase (*ap_2740*, *ap_15570*) and expansin (*ap_2687*). In addition, it was observed that bluish trees showed a higher number of coding transcripts related to the defense response such as jacalin-type lectin domain-containing protein (*ap_18349*), and chitinases (*ap_10285*, *ap_21337*), as well as signal transduction, including many leucine-rich repeat (LRR) receptor-like serine/threonine-protein kinase transcripts (*ap_12440*, *ap_13212*, *ap_57871*, *ap_90651*, *ap_90920*) (Appendix A). In contrast, in green trees, a higher presence of DE transcripts for proteins involved in substrate transport (transporters) such as ammonium transporter (*ap_45727*), formate/nitrite transporter (*ap_1767*), allantoin permease (*ap_17692*), and amino acid permease (*ap_51656*), as well as transcripts encoding members of the major facilitator superfamily involved in the transport of small solutes across cell membranes (*ap_20959*, *ap_34051*, *ap_54447*, *ap_58946*, among others), was observed. Other transcripts differentially expressed in green trees included those encoding different ribosomal proteins (*ap_2470*, *ap_26629*, *ap_31990*, *ap_31976*, *ap_60516*, among others) (Figure 2A), and those involved in carbon metabolism, such as transketolase (*ap_22680*), epimerase (*ap_17677*), methylthioribose-1-phosphate isomerase (*ap_68963*), phosphoenolpyruvate carboxykinase (*ap_31997*), and isocitrate lyase (*ap_27804*), (Appendix A).

Interestingly, a different expression pattern of transcripts involved in chromatin remodeling processes and transcription regulation was also observed (Figure 2A) (Appendix A). Bluish trees exhibited overexpression of transcripts related to histone modification and epigenomic mark deposition, including a SET domain-containing protein (*ap_34496*), histone-lysine N-methyltransferase ASHR1 (*ap_12277*, *ap_24242*) and chromomethylase (CMT) (*ap_5633*) (Figure 2A) (Appendix A). In green trees, upregulation of transcripts was found for the chromatin remodeling process, particularly those related to supporting the interaction between chromatin remodeling proteins and histones, such as Swi3, Ada2, N-Cor, and TFIIIB (SANT) domain-containing protein (*ap_14197*, *ap_16559*, *ap_22702*, *ap_78725*) and transposable elements (TEs) (*ap_21719*, *ap_75667*, *ap_85660*). However, differential expression patterns were observed for transcripts encoding transcription factors (TFs) (Figure 2A) (Appendix A). These differences were also observed in transcripts associated with N metabolism and photosynthesis (Figure 2A).

Singular enrichment analysis (SEA) was performed individually for each gene expression regulation pattern (up- and downregulated) to classify the biological functions of DE transcripts in bluish versus green trees (Appendix A). Only upregulated transcripts were found to be significantly enriched in Gene Ontology (GO) terms in categories under biological processes (BP) and cellular components (CC) (Figure 2B,C). *Cell wall macromolecule metabolic process* (GO:0044036) was a unique term found to be enriched in BP (Figure 2B), while the terms enriched in CC were *extracellular region* (GO:0005576) and *cell wall* (GO:0005618) (Figure 2C).

To validate the transcriptomic analysis, the expression of several selected DE transcripts was analyzed by RT-qPCR (Figure 3) (Appendix A). Within the bluish/green comparison the upregulation of red chlorophyll catabolite reductase (*ap_72669*), RNA recognition motif domain-containing protein (*ap_1400*), chromomethylase-CMT (*ap_5633*) and asparagine synthetase (*ap_14760*) was confirmed, while the downregulation of ribulose bisphosphate carboxylase small subunit (*ap_72505*), SANT domain-containing protein (*ap_22702*) and CYP720B12v2 (*ap_84836*) was also confirmed by quantitative determination of their relative transcript levels.

### 2.3. Physiological Markers Revealed Clear Differences between the Two Phenotypic Variants of Spanish Fir Trees

Several physiological markers were studied to further characterize the phenotypic variants of *A. pinsapo* (bluish/green trees) and validate the transcriptomic data previously obtained (Figure 4). Thus, green trees were observed to have more chlorophyll (both chlorophyll *a* and *b*) than bluish trees (Figure 4A–D). In particular, the chlorophyll *b* content was nearly three times higher in green trees (Figure 4B). In contrast, bluish trees showed a higher content of total carotenoids (Figure 4E) and violaxanthin (Figure 4F). However, no significant changes in antheraxanthin content were observed between green and bluish trees (Figure 4G). Interestingly, one of the major and significant differences in the contents of pigments was the high abundance of chlorophyll *b* in green trees, which contrasted with the higher amounts of total carotenoids exhibited by bluish trees (Figure 4E,F). The contents of proteins and chlorophylls are two suitable markers of N status [28], while cellulose constitutes one of the major sinks for carbon (C) in plants [29]. To study whether there was an alteration in the metabolic status/flux across the two phenotypic variants under study, soluble protein and cellulose content were measured as additional physiological markers. Interestingly, two different patterns were observed. The soluble protein content was significantly higher in green trees (Figure 4H), while the opposite result was obtained for the cellulose content (Figure 4I). Both the C and N contents were higher in green trees (Figure 4J,K). However, the C:N ratio was higher in bluish trees, mainly due to their lower N content (Figure 4L).

### 2.4. Chlorophyll Fluorescence of Needles from Spanish Fir Trees

To further investigate whether the differences in chlorophyll content altered the photochemical performance of PSII, the chlorophyll fluorescence of needles from bluish and green trees was analyzed. Needles from green trees showed a higher PSII maximum quantum efficiency than bluish trees, expressed as Fv/Fm (Figure 5A). Green trees showed higher electron transport rates (ETR) at light intensities above 100 µmol m^−2^ s^−1^ (Figure 5B). Fitting LCs using the Levenberg–Marquardt algorithm [30] rendered a higher photosynthetic efficiency (α) for green branches (Figure 5C). Finally, we found that saturating irradiance (Ek) and maximum electron transport rate (ETRmax) were also higher in green trees (Figure 5D,E).

## 3. Discussion

Transcriptome sequencing is an important starting point in genomic studies in plant species whose genome has not been characterized. This approach is especially useful in conifers that have megagenomes that are approximately six times larger than the human genome. A limited number of nucleotide sequences are available in public databases for Spanish fir [31]; these resources are insufficient to carry out functional and conservation studies. In the present work, a minimum reference transcriptome was defined for *Abies pinsapo* containing 22,769 nonredundant transcripts, which is close to the number of unique transcripts reported for the well-characterized transcriptomes of other conifer species such as *Picea glauca* and *Pinus pinaster* [32,33]. This reference transcriptome also provides a large collection of full-length cDNAs that are required for gene annotation, functional studies, and identification of relevant proteins in *A. pinsapo*.

In the last few years, there has been increasing interest in understanding the ecological roles of intraspecific variation regarding responses to environmental change, especially human-induced alterations to ecosystems and their effects on the loss of biodiversity [8,34,35,36,37]. However, there is still a lack of knowledge on the possible mechanisms underlying intraspecific variation and how to infer the potential ecological impacts of this variation. In the present work, two phenotypic variants of Spanish fir (trees with bluish or green needles) were characterized at the physiological and molecular levels to gain insights into the mechanism of the observed differences. Green trees revealed a higher PSII maximum quantum efficiency (Figure 5) which is consistent with the higher content of chlorophyll (Figure 4), as reported previously in other species [38], and the observed upregulation of transcripts for chlorophyll *a*-*b* binding proteins (Figure 2), the apoproteins of the light-harvesting complex of PSII [39]. In contrast, the lower chlorophyll content and reduced PSII maximum quantum efficiency in bluish trees appear to be closely linked to the upregulation of transcripts encoding red chlorophyll catabolite reductase (Figure 2), an essential enzyme involved in chlorophyll catabolism and degreening [40,41]. These results suggest lower stacking of thylakoids and lower energy transfer efficiency in the chloroplasts of bluish trees compared to green trees [42]. In addition, the higher values of electron transport efficiency (α) observed in needles from green trees (Figure 5C,D) suggest a better capacity for light capture and enhanced photosynthetic efficiency, probably due to more efficient light-harvesting complexes in comparison to bluish trees. Bluish trees appear to have limited electron chain transport and downregulation of transcripts encoding ribulose 1,5-bisphosphate carboxylase/oxygenase (RuBisCO) (Figure 2), the key enzyme of the Calvin cycle [43]. Carotenoids are well known to be accessory pigments that help chlorophyll *a* capture light energy during photosynthesis, maintain the structure and function of photosynthetic complexes, quench chlorophyll triplet states, and play important roles in photoprotection and mitigate antioxidative processes [44,45]. In addition, the lack of carotenes such as violaxanthin, neoxanthin, and lutein has been found to apparently promote a smaller PSII light-harvesting chlorophyll antenna size [46]. Interestingly, bluish trees were found to harbor higher contents of total carotenoids and violaxanthin (Figure 4). Violaxanthin is the precursor of zeaxanthin via the intermediate antheraxanthin [47,48]. The reversible cyclic conversions of violaxanthin, antheraxanthin, and zeaxanthin are called the violaxanthin cycle and have been reported to have profound effects on light harvesting and light energy utilization in PSII [49]. Although it was not possible to quantify zeaxanthin, it is reasonable to hypothesize that bluish trees might harbor higher amounts of zeaxanthin than green trees. Therefore, all these findings suggest that bluish trees tend to compensate for their decreased photosynthetic capacity by modifying the PSII light-harvesting chlorophyll antenna through the accumulation of carotenes as reported in algae [46] and/or reinforcing PSI stability as occurs in terrestrial plants such as tomato [50]. It is possible that through this mechanism, bluish trees could avoid overreduction of the plastoquinone pool and the resulting photoinhibition of the PSII reaction centers [49] to ensure a lower but stable photosynthetic capacity.

It has been widely reported that carbon flux to cellulose production increases with photosynthesis [51]. Surprisingly, a higher cellulose content was found in needles from bluish trees (Figure 4), which agrees with the upregulation of several transcripts encoding cellulose synthase (Figure 2). A priori, this result seems to contradict the previously discussed results. However, mesophyll conductance to CO_2_, one of the main limitations in photosynthesis and thus CO_2_ assimilation [52,53], was recently described to correlate negatively with cellulose and hemicellulose content in the cell walls of conifers [54] due to their thickness [54,55,56]. Therefore, this evidence is consistent with the greater amount of cellulose present in the needles of bluish trees, as well as with the overexpression of genes encoding cellulose synthases and xyloglucan endotransglucosylase/hydrolases.

Regarding carbon (C) metabolism, our results suggest a better C metabolic status in needles from green trees in agreement with the higher photosynthetic capacity observed and the slightly higher total C content (Figure 4). Therefore, our results strongly suggest that the differences observed at the photosynthetic level could alter cell wall composition. Finally, C and N metabolism is well known to be two closely linked metabolic pathways [57]. In the needles from green trees, the upregulation of several transcripts encoding enzymes in the tricarboxylic acid cycle was observed (Figure 2), and this upregulation seems to be closely linked to enhanced N metabolism, as revealed by the higher total N content in needles from green trees and the differences observed related to the C:N ratio (Figure 4). Inorganic N assimilation for amino acid biosynthesis is a metabolic process that requires a continuous supply of energy and carbon skeletons [58,59]. Indeed, our transcriptomic data also suggest a better C and N status in needles from green trees since needles from bluish trees showed an upregulation of transcripts for aspartate aminotransferase (*ap_91455*) and asparagine synthetase (*ap_14760*) (Figure 2). This result is consistent with the lower photosynthetic capacity observed in these trees since the amino acid asparagine is employed as a temporal N reserve when the flux of carbon is limited [60]. In contrast, green trees exhibited a higher N content (Figure 4) and a concomitant upregulation of transcripts encoding nitrate reductase (NR) (Figure 2), strongly supporting active N assimilation since NR catalyzes the rate-limiting step in the nitrate assimilation pathway [61], and is coupled to photosynthetic energy production [62]. Finally, the above findings also agree with the higher content of soluble proteins (Figure 4) and the broad upregulation of ribosomal protein-coding transcripts observed in the needles from green trees (Figure 2).

The above findings suggest that bluish trees may represent a local adaptation or ecotype to stressful climatic (and soil) conditions. The bluish phenotype occurs mainly and is more abundant in the lower ecotone of the *Abies pinsapo* eastern-most populations in the Yunquera forest (personal communication from J.B. López-Quintanilla—Forest Service regional coordinator of the *A. pinsapo* Recovery Plan-). This area shows some of the driest conditions across the whole *A. pinsapo* distribution area [63,64]. Moreover, the Yunquera population showed the earliest and most intense decline and dieback process associated with ongoing climate change at the regional level [65]. In a comparative study using most of the circum-Mediterranean fir species, the fir populations currently suffering under warmer and drier conditions were predicted to be subjected to a very high level of vulnerability by the end of the XXI century, with the *Abies pinsapo* population at Yunquera-Sierra de las Nieves being one of the most affected among the 30 populations studied around the whole Mediterranean Basin [66]. On the other hand, low soil P availability and a N/P co-limitation of productivity have been reported for the Yunquera forest [67,68]. Thus, our results support that the bluish phenotype might be useful for generating a variety that is especially well adapted to stressful conditions in climate-adaptation plans for this species (e.g., ex situ plantations, strengthening the adaptive capacity of forest masses in areas with suboptimal climatic and edaphic conditions).

Moreover, the obtained results also lead to the question of the causes behind these differences. In this case, it is tempting to hypothesize that the observed differences could be, at least, partially attributed to epigenomic changes, as suggested by the transcriptomic analyses. Therefore, clearly different tendencies related to chromatin structure-related transcripts were observed (Figure 2). For example, in needles from green trees, upregulation of coding transcripts for several SANT domain-containing proteins was observed (Figure 2). This protein domain is widely known to allow the interaction of chromatin remodeling proteins with histones [69,70,71]. On the opposite side of the comparison, the upregulation of several histone-lysine N-methyltransferase ASHR1 and DNA methyltransferase chromomethylase (*CMT*) coding transcripts was observed (Figure 2). These proteins have been widely reported to mediate transcription through posttranslational histone modifications [72] and cytosine methylation of specific DNA sequence [73] processes. Therefore, different chromatin states and genome DNA methylation patterns could affect gene transcription, as has been described in multiple eukaryotic organisms [74,75,76,77,78], since these modifications play important roles in the regulation of chromatin condensation and DNA accessibility [79]. All these phenomena could also explain the differences observed in the differential expression of several transcription factors (Figure 2) that could be involved in the differential gene responses observed. Future works on epigenomic marks in Spanish fir, along with genome information, will be needed to elucidate this point.

## 4. Conclusions

Knowledge of the molecular basis of growth and development as well as the mechanisms underlying the response to environmental stimuli is essential for the conservation of forest trees. The Spanish fir tree is an under-investigated tree in molecular terms, and the current knowledge of basic biological processes is quite limited. However, advances in high-throughput techniques are enabling holistic approaches for a better understanding of fundamental tree biology. In this work, the transcriptome of Spanish fir was assembled, providing a large catalog of expressed genes (22,769), within which a high proportion of them (approximately 55%) were full-length transcripts. This achievement represents a valuable resource for functional genomics studies in this relict conifer species and for gene annotation of its genome. As the first case of study for the implementation of the above genomic resource, two phenotypic variants of Spanish fir trees were analyzed at the physiological and molecular levels. Differential regulation of the transcriptome was observed in the needles from bluish and green trees with changes in the light-harvesting complex/photosynthetic capacity and cell wall composition. These findings suggest a local adaptation for the differential management of photosynthetic efficiency and the use of nutrients such as N. With regard to the potential mechanism behind this intraspecific variation, a hypothesis is proposed based on possible changes in the chromatin state and genomic DNA methylation patterns since epigenetic changes may be inherited by the next generation. Further research efforts are needed to provide a better understanding of the nature of these trees to design appropriate conservation and management strategies for these endangered tree species.

## 5. Materials and Methods

### 5.1. Plant Material

Plant material used for transcriptome sequencing was harvested at different locations in *Parque Nacional de la Sierra de las Nieves* (Yunquera, Málaga, Spain). A variety of *A. pinsapo* tissues were collected in April 2013 from adult trees at *Puerto Saucillo* and seedlings for cDNA synthesis and pyrosequencing. Additionally, as a source of RNA for sequencing analysis using the Illumina platform, in the dry summer (august) and wet spring (may) seasons of 2017, we collected needles samples from trees in dense stands, trees in thinned stands, and artificially shadowed trees within thinned stands. For this, we used control and treated plots from a climate-adaptation field assay at *Cañada Bellina* in which structural diversity enhancement thinning treatments had been applied in 2004 to declining and dense *A. pinsapo* stands in order to reduce intraspecific competition for water and light (see [80,81] for more details). Needles from adult trees with green and bluish leaf phenotypes were sampled at *Hoyo Millán* at 1080 m altitude. The harvesting of needles from six independent green and bluish trees was carried out in April 2021. The samplings were carried out 5–6 h after sunrise. Harvested samples were immediately frozen in liquid N and subsequently placed in dry ice for transport to the laboratory. All frozen samples were homogenized with a Mixer Mills MM400 (Retsh, Haan, Germany) and stored at −80 °C until RNA isolation or further use.

### 5.2. Total RNA Isolation

RNA was extracted as described by [33] from ground powder stored at −80 °C. A treatment with RQ1 RNase-Free DNase (Promega, Madison, WI, USA) was applied to remove genomic DNA from the RNA samples. Total RNA quantification and purity were estimated using a NanoDrop ND-1000 spectrophotometer (Thermo Scientific, Waltham, MA, USA), and RNA integrity was checked by agarose gel. RNA quality was also determined in a 5200 Fragment Analyzer System (Agilent, Santa Clara, CA, USA) using the Agilent DNF-472 HS RNA Kit (Agilent, Santa Clara, CA, USA). Only samples with RIN > 7 were selected for sequencing.

### 5.3. Whole Transcriptome Sequencing and Transcriptome Assembly

Transcriptome sequencing using the GS-FLX+ platform was performed at the Universidad de Málaga ultrasequencing facility [82], and the sequencing by Illumina was carried out by Novogen (HK). RNA libraries for sequencing were prepared using a GS-FLX Titanium kit (Roche Applied Sciences, IN, USA) and a TruSeq RNA kit (Illumina, San Diego, CA, USA) according to the manufacturer’s instructions. The samples were sequenced using GS-FLX as reads of 1000 bp in length and Illumina NovaSeq 6000 as paired-end reads of 150 bp in length. Data have been deposited in the NCBI’s SRA database [83] and are accessible in the BioProjects PRJNA781601 and PRJNA858095, respectively.

Subsequent read processing was made in the Picasso supercomputer at Supercomputing and Bioinnovation Center (SCBI, Universidad de Málaga). The raw reads were trimmed (quality and contamination) using SeqTrimBB software (https://github.com/rafnunser/seqtrimbb, accessed on 10 January 2020)). Only the pairs in which both reads passed the quality test were further analyzed (Q > 20). The resultant reads were assembled using Trinity 2.11 [84]. Contigs lower than 400 pb were eliminated. For the rest of contigs, the redundancy was reduced using CD-HIT-EST software [85]. The final transcriptome of *A. pinsapo* was used as the reference for the read mapping that was performed with BWA using the MEM option [86]. The read count was obtained with the phyton script *sam2counts* (https://github.com/vsbuffalo/sam2counts, accesed on 25 March 2021). In the transcriptomic profiling of bluish and green needles, differentially expressed (DE) transcripts were identified using the edgeR package for R. The transcripts were normalized by cpm and filtered; 2 cpm in at least 2 samples [87]. Each sample was from a single tree exhibiting a bluish or a green phenotype. Samples were grouped by phenotypic color variation (Appendix A). Only the transcripts with |logFC| > 1 and FDR < 0.05 were considered as differentially expressed (DE). These RNA-seq data have been deposited in the NCBI’s Gene Expression Omnibus [88] and are accessible through GEO Series with the accession number GSE189122 (https://www.ncbi.nlm.nih.gov/geo/query/acc.cgi?acc=GSE189122 accessed on 11 July 2022).

### 5.4. Transcriptome Assembly Analysis

The assembled transcriptome of *A. pinsapo* was analyzed using Full-LengtherNext (FLN; https://rubygems.org/gems/full_lengther_next/versions/1.0.1 accessed on 11 July 2022) software to provide gene description and to identify which unigenes corresponded to full-length cDNAs (FLcDNAs), noncoding RNAs (ncRNAs), and putative protein sequence, and to obtain a quick preview of unigene content in the Spanish fir transcriptome.

### 5.5. Functional Annotation and Enrichment Analyses

The assembled transcriptome was functionally annotated with BLAST2GO [89] using DIAMOND software with blastx option [90] against the NCBI’s plants-nr database [83]. Blast results were considered valid with e < 1.0 × 10^−6^. Singular enrichment analysis (SEA) of the GO terms was made in the AGRIGO v2.0 web tool under standard parameters using as GO term reference the whole assembled transcriptome annotation [91].

### 5.6. Reverse-Transcription Quantitative PCR (RT-QPCR)

Reverse-transcription reactions were performed using iScrptTM Reverse-Transcription Supermix (Bio-Rad, Hercules, CA, USA) using 1 µg of total RNA. The qPCR reactions were carried out using 5 ng of cDNA and qPCRBIO SyGreen Mix Lo-ROX (PCR BIOSYSTEMS, London, UK) in a final volume of 10 µL. The reactions were developed on a C1000TM Thermal Cycler with a CFX384TM Touch Real-Time PCR Detection System (Bio-Rad, Hercules, CA, USA) under the following conditions: 3 min at 95 °C (1 cycle), 1 s at 95 °C, and 5 s at 60 °C (50 cycles), with a melting curve from 60 °C to 95 °C. The raw fluorescence data from each reaction were fitted to the MAK3 model [92]. The initial target concentration (D0 parameter) was determined using the R package qpcR [93]. To normalize expression data, two ortholog reference genes previously tested for RT-qPCR experiments in maritime pine [94] were identified and tested: PSFP (peptidase S24/S26A/S26B/S26C family protein) and SLAP (Saposin-like aspartyl protease). For RT-qPCR analysis, six biological replicates and three technical replicates per sample were used. Primers used for RT-qPCR are presented in Appendix A.

### 5.7. Soluble Proteins, Chlorophyll and Carotenoid Determination

Soluble proteins were extracted using 50 mg in the case of needles. The extraction was performed by adding 1 mL of extraction buffer (50 mM Tris-HCl pH 8, 1 mM EDTA, 10 mM MgCl_2_, 0.5 mM dithiothreitol (DTT), 20% (*w*/*v*) glycerol, 0.1% (*v*/*v*) Triton X-100, 1% (*w*/*v*) polyvinylpyrrolidone (PVP), and 1% (*w*/*v*) polyvinyl(poly)pyrrolidone (PVPP)) and 30 mg of fine sea sand. The resulting extract was centrifuged at 16,000× *g* for 30 min at 4 °C. The obtained supernatants were recovered and used for protein determination through Bradford’s procedure using a commercial reagent (Protein Assay Dye Reagent; Bio-Rad, Hercules, CA, USA) and bovine serum albumin (BSA) as a standard [95].

The quantification of chlorophylls and total carotenoids (xanthophylls and carotenes) was performed using 50 µL of protein extract mixed with 950 µL of 80% (*v*/*v*) acetone. Samples were incubated at 4 °C overnight. The resulting extracts were centrifuged at 13,500× *g* at 4 °C for 10 min, and the absorbance was measured at 470, 664, and 647 nm. The contents of chlorophylls and total carotenoids were calculated according to [96].

### 5.8. Pigment Extraction and Quantification

Pigment quantification was determined using 10–15 mg of grounded needles from blue and green trees. Samples were mixed with 500 µL of N, N-dimethylformamide, and they were extracted in the dark overnight at 4 °C. After the incubation period, samples were filtered using a 0.2 µm PTFE syringe filter and placed in amber vials with inserts. Vials containing samples were placed in a thermos-tatted autosampler, and they were injected into the HPLC system (HPLC Ultimate 3000, Thermo Fisher Scientific, Waltham, MA, USA) equipped with ACE Excel 2 C18 columns (100 × 3 mm) and carried by the mobile phase to UV-Vis Dionex Ultimate 3000 spectrophotometer (Thermo Fisher Scientific, Waltham, MA, USA). The mobile phase composition was in a gradient (0–8 min 75% A and 25% B, 8–10 min 25% A and 75% B, 10–18 min 10% A and 90% B, 18–40 min 100% B, 40–50 min 75% A and 25% B and 50–61.2 min 75% A and 25% B), where A was 0.05 M tetrabutylammonium, 1 M ammonium acetate in H_2_O, and B was methanol/acetone (50:50 *v*/*v*). The flow rate used was 0.3 mL/min. The equipment software (Xcalibur, Thermo Fisher Scientific) was used to process the chromatography data to quantify the amounts of carotenoids present in the samples. Certified standards were used as references.

### 5.9. Cellulose Quantification

Cellulose content was determined by the anthrone method [97]. Briefly, 100 mg of tissue powder was boiled in 1 mL acetic-nitric reagent (acetic acid:nitric acid:water, 8:1:2) for 30 min to remove hemicellulosic and lignin carbohydrates. After centrifugation at 4500× *g* for 20 min, the supernatant was removed, and the remaining material was washed twice with 1 mL of distilled water. The cellulose samples were then hydrolyzed in 67% (*v*/*v*) sulfuric acid for 16 h at 37 °C, and the glucose content of the samples was determined as follows: 100 µL of the sulfuric acid hydrolyzed samples was mixed with 900 µL of water and 10 µL of 0.2% (*w*/*v*) anthrone in concentrated sulfuric acid on ice. The samples were boiled for 10 min, and then the absorbance was measured at 630 nm. Cellulose content was calculated based on a previously determined standard using commercial cellulose from Sigma-Aldrich (St. Louis, MO, USA).

### 5.10. Elemental Analysis

Ground powder (100 mg) of six bluish and green needles samples were dried at 80 °C for 48 h in an oven. Total carbon (C) and nitrogen (N) contents were determined by an elemental macro-analyzer Leco truSpec CHNS (Leco Corporation, St. Joseph, MI, USA) at the Atomic Spectrometry Unit, University of Málaga.

### 5.11. Chlorophyll Fluorescence Analysis

Variable chlorophyll A fluorescence was monitored using a JUNIOR-PAM chlorophyll fluorometer (Heinz Walz GmbH, Effeltrich, Germany) controlled by WinControl-3 software (Heinz Walz GmbH, Effeltrich, Germany). Three branches were harvested from nine independent green and bluish trees. For measurements, after 20 min dark-incubation, the fluorometer glass fiber was set at 1 mm distance from the needle using the magnetic leaf clip. Once stable minimum fluorescence (Fo) was detected, the rapid light curve program protocol was started. Twenty seconds steps of actinic illumination (PAR) ranging from 0 to 420 µmol m^−2^ s^−1^ (PPFD) were applied. The maximum photochemical quantum yield of photosystem II (PSII) was recorded at the first saturating pulse and relative electron transport rate (ETR), calculated for each PAR illumination by WinControl-3. Light Curves were fitted using Levenberg–Marquadt algorithm [98] (ETR = ETRmax · tanh (α·PPFD/ETRmax)).

## Figures and Tables

**Figure 1 ijms-23-09351-f001:**
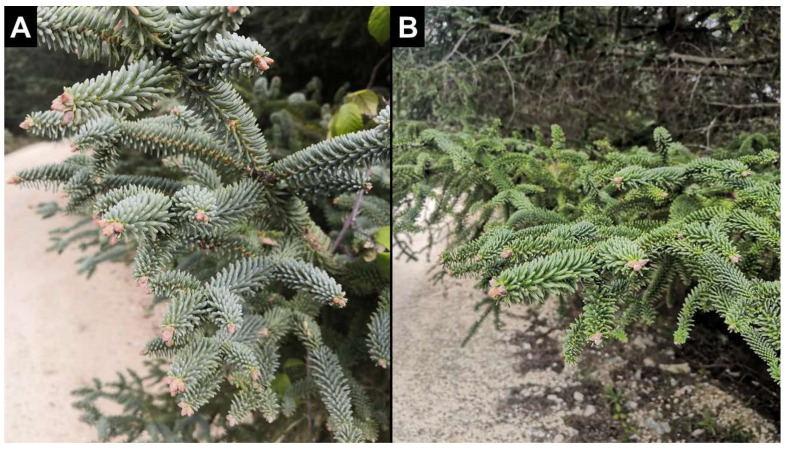
Image of the bluish (**A**) and green (**B**) needles of adult Spanish fir.

**Figure 2 ijms-23-09351-f002:**
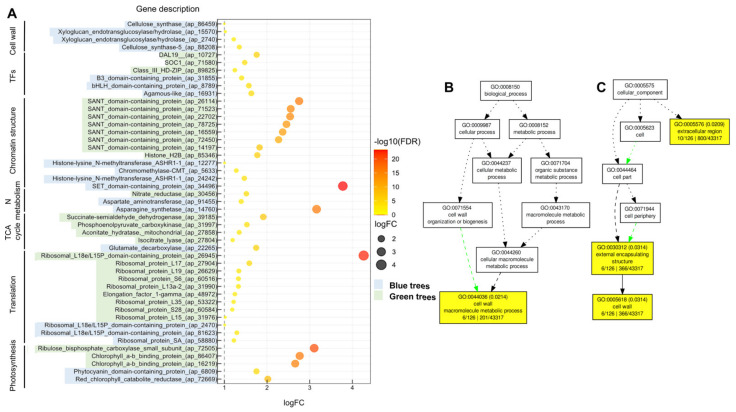
Transcriptome analysis on the needles from bluish and green trees. (**A**) Graphical representation of differentially expressed (DE) transcripts based on their representation (logFC) and statistical significance (−log10 FDR) regarding different cellular processes/pathways. Singular enrichment analysis (SEA) of the GO terms was made in the AGRIGO v2.0 web tool under standard parameters using as GO term reference the whole assembled transcriptome annotation. Blue-labeled transcripts correspond to a higher abundance of DE transcripts in needles from bluish trees. Green-labeled transcripts correspond to a higher abundance of DE transcripts in needles from green trees. (**B**) Significant GO biological process terms from significant DE transcripts after a SEA analysis. (**C**): Significant GO cellular component terms from significant DE transcripts after a SEA analysis. The harvesting of needles from six independent green and bluish trees was carried out in April 2021 at *Hoyo Millán* at 1080 m altitude.

**Figure 3 ijms-23-09351-f003:**
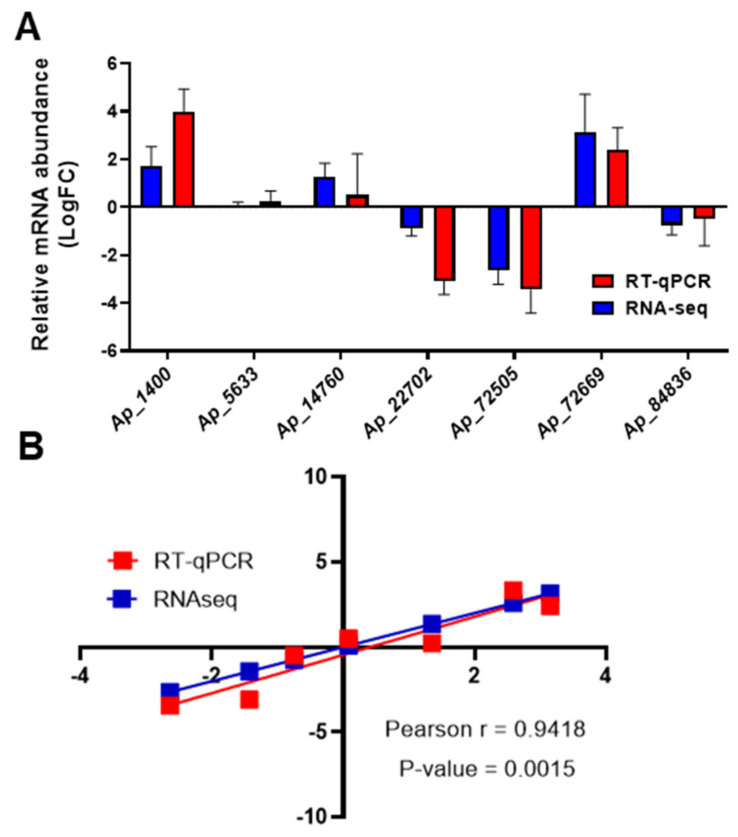
Validation of transcriptome sequencing. (**A**) Experimental validation of RNA-seq results for 7 DE transcripts through RT-qPCR (quantitative reverse-transcription polymerase chain reaction). *Ap_1400*: Putative RNA recognition motif domain-containing protein; *Ap_5633*: Chromomethylase (CMT); *Ap_14760*: Asparagine synthetase; *Ap_22720*: SANT domain-containing protein; *Ap_72505*: Ribulose bisphosphate carboxylase (RuBisCO) small subunit; Ap_72669: Red chlorophyll catabolite reductase; *Ap_84836*: Cytochrome P450 CYP720B12v2. Red and blue columns correspond to transcript abundance quantified by RT-qPCR and RNA-seq, respectively. (**B**) Gene expression correlation between RT-qPCR and RNA-seq data. Pearson r: Pearson correlation coefficient.

**Figure 4 ijms-23-09351-f004:**
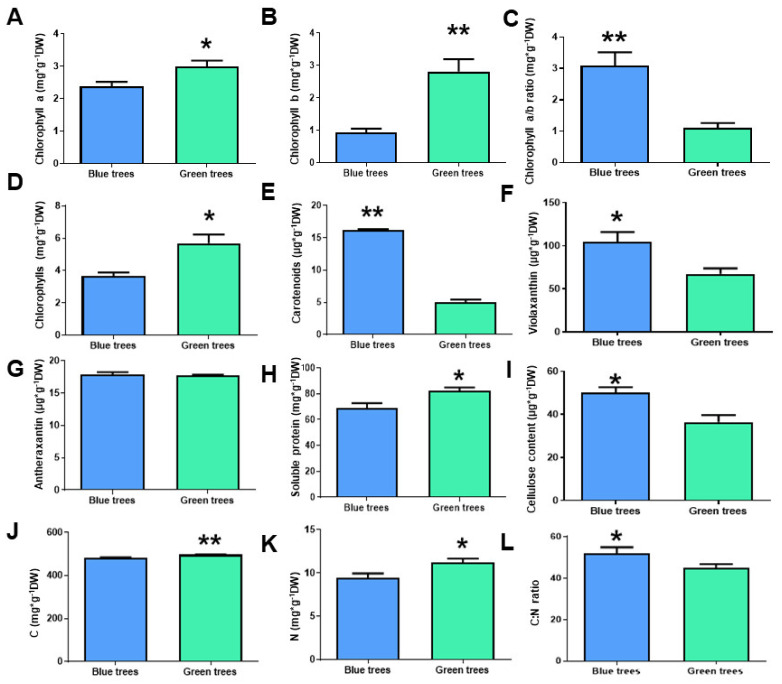
Profiles of plant physiological markers in needles from bluish and green trees. (**A**): Chlorophyll A content. (**B**): Chlorophyll B content. (**C**): Chlorophyll a/b ratio. (**D**): Total chlorophyll content. (**E**): Total carotenoid content. (**F**): Violaxanthin content. (**G**): Antheraxanthin content. (**H**): Soluble protein content. (**I**): Cellulose content. (**J**): Carbon (C) content. (**K**): Nitrogen (N) content. (**L**): C:N ratio. Blue columns correspond to needles from bluish trees. Green columns correspond to needles from green trees. Significant differences were determined with a *t*-test for each phenotypic needle variation. Asterisks (*) above the columns show significant differences based on: * at *p* < 0.05; ** at *p* < 0.01, Error bars show SE with n = 6.

**Figure 5 ijms-23-09351-f005:**
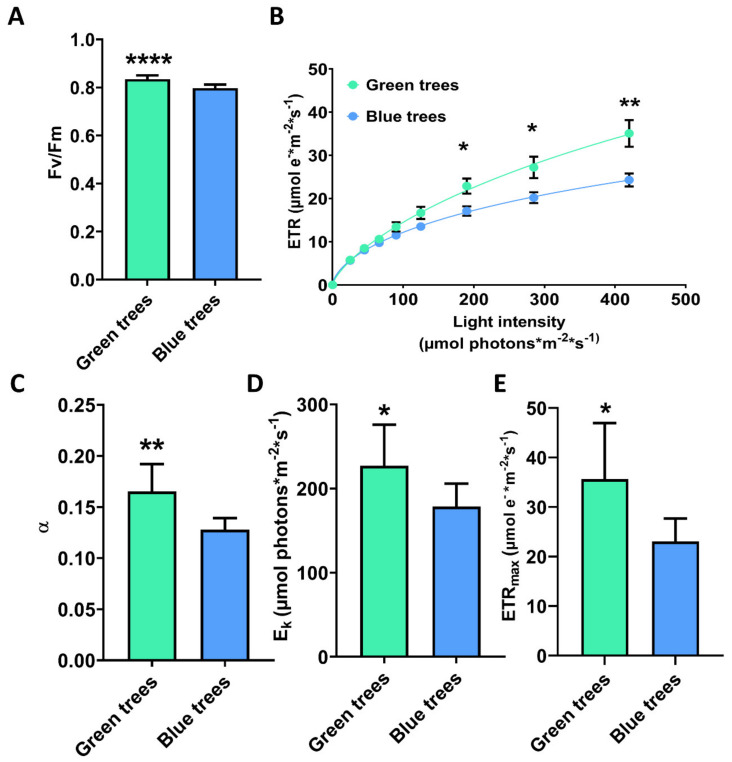
Photosynthesis parameters based on chlorophyll fluorescence analysis. (**A**) Maximum quantum efficiency of photosysytem II (PSII) photochemistry (Fv/Fm). (**B**) Relative electron transport rate (ETR). (**C**) Photosynthetic efficiency (α). (**D**) Light saturation index (E_k_). (**E**) Maximum electron transport rate (ETR_max_). Blue columns correspond to needles from bluish trees. Green columns correspond to needles from green trees. Significant differences were determined with a *t*-test for each phenotypic needle variation. Asterisks (*) above the columns show significant differences based on: * at *p* < 0.05; ** at *p* < 0.01, **** at *p* < 0.0001. Error bars show SE with n = 9.

**Table 1 ijms-23-09351-t001:** Description of plant material used for transcriptome analysis.

Sequencing Platform	Type of Plant Material	Experimental Conditions	Place	Sampling Time
454	Needles	50-year-old tree	Puerto Saucillo	April 2013
454	Branch	50-year-old tree	Puerto Saucillo	April 2013
454	Male cones	50-year-old tree	Puerto Saucillo	April 2013
454	Female cones	50-year-old tree	Puerto Saucillo	April 2013
454	Roots	50-year-old tree	Puerto Saucillo	April 2013
454	Needles	One-year-old seedlings	Puerto Saucillo	April 2013
454	Stem	One-year-old seedlings	Puerto Saucillo	April 2013
454	Roots	One-year-old seedlings	Puerto Saucillo	April 2013
Illumina	Needles	Adult trees in dense unthinned stands (high competition for light and water)	Cañada Bellina	May and August 2017
Illumina	Needles	Adult trees in thinned stands aiming structural diversity enhancement (low competition for water and light)	Puerto Saucillo-Llano del Alcornicalejo	May and August 2017
Illumina	Needles	Artificially shadowed (to mimic the light environment in dense stands) adult trees within thinned stands (low competition for water but high for light)	Puerto Saucillo-Llano del Alcornicalejo	May and August 2017
Illumina	Needles	Green tree (30 years old)	Hoyo Millán	April 2021
Illumina	Needles	Bluish tree (30 years old)	Hoyo Millán	April 2021

**Table 2 ijms-23-09351-t002:** Summary of final data for transcriptome of *A. pinsapo*.

	Absolut Number	%
**Input (total unigenes)**	**76,481**	**100.00**
**Artifacts ^1^**	6670	8.29
*Unmapped transcripts*	0	0.00
*Misassembled*	131	1.96
*Chimeras*	6307	94.56
**Unigenes after resolving artifacts**	**73,798**	**91.71**
*Unigenes > 500 pb*	61,840	83.80
*Unigenes > 200 pb*	73,695	99.86
*N50 (bp)*	1799	-
*N90 (bp)*	599	-
**Unigenes with ortholog ^1^**	**36,503**	**49.46**
*Different ortholog IDs*	19,086	52.29
*Complete transcripts (full-length)*	20,135	55.16
*Different complete transcripts*	12,545	34.37
**Unigenes without ortholog ^1^**	**37,021**	**50.17**
* **Coding (all)** *	12,745	34.43
*Coding > 200 bp*	12,745	34.43
*Coding > 500 bp*	11,077	29.92
*Putative coding*	9336	73.25
*Nonredundant coding*	3409	26.74
* **Unknown (all)** *	24,276	65.57
*Unknown > 200 bp*	24,204	65.38
*Unknown > 500 bp*	17,356	46.88
**Putative ncRNA**	**274**	**0.37**

^1^ Percents for subclassifications of this category were calculated using this line as 100% reference. letters in bold indicate a general category of transcripts.

## Data Availability

Not applicable.

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
