# Peer review of "Transcriptome Analysis and Intraspecific Variation in Spanish Fir (Abies pinsapo Boiss.)"

_ijms, 2022, doi:10.3390/ijms23169351_

Round 1

Reviewer 1 Report

This is an interesting and valuable study describing a molecular differences between bluish and green Spanish firs growing at multistress conditions. Authors performed analysis of high-throughput cDNA sequencing and transcriptome assembly to reveal differently regulated genes. The results have been validated with quantitative PCR and by changes in several variables: chlorophylls, carotenoids, cellulose, carbon and nitrogen. However, the ms suffers from several weaknesses, especially in the part describing photosynthetic parameters and in the entire Discussion.

Specific points:

English should be carefully checked and improved throughout the text.

In my opinion, it would be more clear to move the Table 1 and Fig.1 to the beginning of Results or to the end of Introduction.

Considering a variation of conditions described in M & M (p.11.l.390-392), namely wet spring, dry summer, dense and thinner stand, it should be precisely described in the legend to Fig.2. – for which conditions these data are representative for?

The abbreviation “bp” currently has two meanings in the ms “base pair” (p.3, Table 2) and “biological process” (p.4). Obviously, both of them are proper, but the abbreviation BP on p.4.l.161 can easily be removed without any consequences.

Fig.4: On the basis of changes in the contents of chl a and b - a dramatic change in the chl a/b ratio can be calculated. For bluish needles I calculated value of about 2.5 (quite typical), whereas, for green needles values of ab 1, which is very untypical and similar to the transgenic Arabidopsis plants overexpressing chlorophyll a oxygenase that lacks the regulatory domain and has a high overproduction of chl b (see paper of Tanaka and Tanaka 2011: BBA doi:10.1016/j.bbabio.2011.01.002). Therefore, instead of the total chlorophylls on Fig. 4C it would be more exciting to show the ratios of chl a/b. Next, to discuss it deeply in the Discussion section (searching for any data on conifers).

Fig.5A: Four stars indicating the significance on panel A must be a mistake.

Fig. 5B: The light curves of ETR are most likely a “rapid light curves”, which means that each step of actinic light intensity is about 1 min or less (this information should be included in M & M). Whereas, the longer light curves (4-5 min per each step of actinic light) would be more trustful. Hence, these data are very preliminary.

Fig. 5C-E: These data are redundant and they are just the results of numerical analysis made on the data shown on Fig. 5B. According to my knowledge, the algorithm of Levenberg-Marquardt (published in 1978) used for this numerical analysis has not been verified in biological samples. While the biological system, such as photosynthetic electron transport, is more complex and is regulated by several mechanisms which are activated at different stages of the light stress. Hence, it would be much more convincing to extend the measuring program up to the light intensities causing ETR saturation, ETRmax or PSII photoinhibition, instead of extrapolating these points. Further on, the parameter α (Fig. 5C) and ETRmax (Fig. 5E) and are the components of the same equation hence I do not see much sense in showing them both in separate figures. It would be sufficient to show ETRmax, however with a clear indication that these are “predicted data”. Moreover, α is merely a coefficient which allows for fitting of measured ETR values to the curve therefore I cannot agree for giving it the name “photosynthetic efficiency”. This suggests a more general information about photosynthesis which is untrue. Presentation of calculated values of Ek (panel D), does not bring to any better understanding of differences in PSII quantum efficiency between the two type of needles, therefore I would skip it.

Description of the 0Y axis in all panels of Fig. 5 should be unified. The names of parameters should be either named in full or abbreviated.

Discussion of results in general is too superficial.

Sentence in l.255-258, with correlation mentioned therein, is not very informative.

One cannot agree with the sentence (l.258-259) that a higher chlorophyll content results in enhanced Fv/Fm. There is not so such straightforward relation between these variables. Moreover, the paper of Sharma et al (2015), cited here, is not present in references.

The suggestion about changes in thylakoid stacking is too speculative (l.265).

The conclusion given on l. 268 that high ETR values is probably due to a richer pigment composition is not proper.

The information about loroxanthin in the Discussion (l.276) is confusing. This pigment has been detected so far in algae (van den Berg and Croce 2022: Front Plant Sci, DOI: 10.3389/fpls.2022.797294). Also, the speculation that carotenoids increase the PSII antenna size is misleading and the literature cited there comes only from the study on algae.

The sentence describing the size of PSII antenna and PSI stability (l.282-290) is far too speculative in relation to here obtained results.

The conclusion linking the changes in N metabolism with changes in transcripts of TCA cycle enzymes does also come too far (l.308-309).

In regard to the discussion of the data dealing with PSII quantum efficiency I suggest to add more information about the protective function of PSII photoinhibition which might be the case in bluish needles.

Discussion of transcriptomic data concerning the chromatin structure and modifications (last paragraph of Discussion) fits better to the part describing changes in transcriptome (at the beginning of the Discussion).

Abstract should be shorten and made more concise.

This is an interesting and valuable study describing a molecular differences between bluish and green Spanish firs growing at multistress conditions. Authors performed analysis of high-throughput cDNA sequencing and transcriptome assembly to reveal differently regulated genes. The results have been validated with quantitative PCR and by changes in several variables: chlorophylls, carotenoids, cellulose, carbon and nitrogen. However, the ms suffers from several weaknesses, especially in the part describing photosynthetic parameters and in the entire Discussion.

Specific points:

English should be carefully checked and improved throughout the text.

In my opinion, it would be more clear to move the Table 1 and Fig.1 to the beginning of Results or to the end of Introduction.

Considering a variation of conditions described in M & M (p.11.l.390-392), namely wet spring, dry summer, dense and thinner stand, it should be precisely described in the legend to Fig.2. – for which conditions these data are representative for?

The abbreviation “bp” currently has two meanings in the ms “base pair” (p.3, Table 2) and “biological process” (p.4). Obviously, both of them are proper, but the abbreviation BP on p.4.l.161 can easily be removed without any consequences.

Fig.4: On the basis of changes in the contents of chl a and b - a dramatic change in the chl a/b ratio can be calculated. For bluish needles I calculated value of about 2.5 (quite typical), whereas, for green needles values of ab 1, which is very untypical and similar to the transgenic Arabidopsis plants overexpressing chlorophyll a oxygenase that lacks the regulatory domain and has a high overproduction of chl b (see paper of Tanaka and Tanaka 2011: BBA doi:10.1016/j.bbabio.2011.01.002). Therefore, instead of the total chlorophylls on Fig. 4C it would be more exciting to show the ratios of chl a/b. Next, to discuss it deeply in the Discussion section (searching for any data on conifers).

Fig.5A: Four stars indicating the significance on panel A must be a mistake.

Fig. 5B: The light curves of ETR are most likely a “rapid light curves”, which means that each step of actinic light intensity is about 1 min or less (this information should be included in M & M). Whereas, the longer light curves (4-5 min per each step of actinic light) would be more trustful. Hence, these data are very preliminary.

Fig. 5C-E: These data are redundant and they are just the results of numerical analysis made on the data shown on Fig. 5B. According to my knowledge, the algorithm of Levenberg-Marquardt (published in 1978) used for this numerical analysis has not been verified in biological samples. While the biological system, such as photosynthetic electron transport, is more complex and is regulated by several mechanisms which are activated at different stages of the light stress. Hence, it would be much more convincing to extend the measuring program up to the light intensities causing ETR saturation, ETRmax or PSII photoinhibition, instead of extrapolating these points. Further on, the parameter α (Fig. 5C) and ETRmax (Fig. 5E) and are the components of the same equation hence I do not see much sense in showing them both in separate figures. It would be sufficient to show ETRmax, however with a clear indication that these are “predicted data”. Moreover, α is merely a coefficient which allows for fitting of measured ETR values to the curve therefore I cannot agree for giving it the name “photosynthetic efficiency”. This suggests a more general information about photosynthesis which is untrue. Presentation of calculated values of Ek (panel D), does not bring to any better understanding of differences in PSII quantum efficiency between the two type of needles, therefore I would skip it.

Description of the 0Y axis in all panels of Fig. 5 should be unified. The names of parameters should be either named in full or abbreviated.

Discussion of results in general is too superficial.

Sentence in l.255-258, with correlation mentioned therein, is not very informative.

One cannot agree with the sentence (l.258-259) that a higher chlorophyll content results in enhanced Fv/Fm. There is not so such straightforward relation between these variables. Moreover, the paper of Sharma et al (2015), cited here, is not present in references.

The suggestion about changes in thylakoid stacking is too speculative (l.265).

The conclusion given on l. 268 that high ETR values is probably due to a richer pigment composition is not proper.

The information about loroxanthin in the Discussion (l.276) is confusing. This pigment has been detected so far in algae (van den Berg and Croce 2022: Front Plant Sci, DOI: 10.3389/fpls.2022.797294). Also, the speculation that carotenoids increase the PSII antenna size is misleading and the literature cited there comes only from the study on algae.

The sentence describing the size of PSII antenna and PSI stability (l.282-290) is far too speculative in relation to here obtained results.

The conclusion linking the changes in N metabolism with changes in transcripts of TCA cycle enzymes does also come too far (l.308-309).

In regard to the discussion of the data dealing with PSII quantum efficiency I suggest to add more information about the protective function of PSII photoinhibition which might be the case in bluish needles.

Discussion of transcriptomic data concerning the chromatin structure and modifications (last paragraph of Discussion) fits better to the part describing changes in transcriptome (at the beginning of the Discussion).

Abstract should be shorten and made more concise.

Author Response

Reviewer 1:

Comments and Suggestions for Authors

This is an interesting and valuable study describing a molecular differences between bluish and green Spanish firs growing at multistress conditions. Authors performed analysis of high-throughput cDNA sequencing and transcriptome assembly to reveal differently regulated genes. The results have been validated with quantitative PCR and by changes in several variables: chlorophylls, carotenoids, cellulose, carbon and nitrogen. However, the ms suffers from several weaknesses, especially in the part describing photosynthetic parameters and in the entire Discussion.

Thank you for the critical evaluation of our manuscript

Specific points:

English should be carefully checked and improved throughout the text.

English language has been carefully checked and edited throughout the text. Editing certificate is attached.

In my opinion, it would be more clear to move the Table 1 and Fig.1 to the beginning of Results or to the end of Introduction.

Agreed. The text has been modified according to the reviewer suggestion.

Considering a variation of conditions described in M & M (p.11.l.390-392), namely wet spring, dry summer, dense and thinner stand, it should be precisely described in the legend to Fig.2. – for which conditions these data are representative for?

Thank you for this comment. The sampling in a variety of conditions was considered for the sequencing of the whole transcriptome. The RNA-seq of needles from blueish and green trees was specifically performed in the conditions described in the legend of Figure 2.  The title of figure 2 has been modified for a better understanding of the experiments.

The abbreviation “bp” currently has two meanings in the ms “base pair” (p.3, Table 2) and “biological process” (p.4). Obviously, both of them are proper, but the abbreviation BP on p.4.l.161 can easily be removed without any consequences.

As nicely stated by the reviewer, the abbreviation "bp" for base pair is widely used in molecular biology meanwhile BP is used to abbreviate biological processes in which a particular transcript is enriched.  We believe that the distinction in lower and uppercase in their specific contexts is suffcient for the proper understanding of data.   

Fig.4: On the basis of changes in the contents of chl a and b - a dramatic change in the chl a/b ratio can be calculated. For bluish needles I calculated value of about 2.5 (quite typical), whereas, for green needles values of ab 1, which is very untypical and similar to the transgenic Arabidopsis plants overexpressing chlorophyll a oxygenase that lacks the regulatory domain and has a high overproduction of chl b (see paper of Tanaka and Tanaka 2011: BBA doi:10.1016/j.bbabio.2011.01.002). Therefore, instead of the total chlorophylls on Fig. 4C it would be more exciting to show the ratios of chl a/b. Next, to discuss it deeply in the Discussion section (searching for any data on conifers).

Agreed. Following the reviewer suggestion we have modified Figure 4 to show the ratios of chla/b. However, values exhibited by green trees are not untypical for conifers. Furthermore, we believe that the observed differences in pigment contents are apropriately discussed in the manuscript. The  higher content of both chlorophyll a and chlorophyll b of green trees is consistent with  the observed changes in chlorophyll fluorescence and the upregulation of transcripts for chlorophyll a-b binding proteins. In contrast, the results found in bluish trees suggest changes in the organization of tylakoids and lower energy transfer effciency in their chloroplasts.

Fig.5A: Four stars indicating the significance on panel A must be a mistake.

Thank you for the comment. We have revised the significance of data and verified that everything is fine. A summary of the statistical analysis is available upon request.

Fig. 5B: The light curves of ETR are most likely a “rapid light curves”, which means that each step of actinic light intensity is about 1 min or less (this information should be included in M & M). Whereas, the longer light curves (4-5 min per each step of actinic light) would be more trustful. Hence, these data are very preliminary.

Fig. 5C-E: These data are redundant and they are just the results of numerical analysis made on the data shown on Fig. 5B. According to my knowledge, the algorithm of Levenberg-Marquardt (published in 1978) used for this numerical analysis has not been verified in biological samples. While the biological system, such as photosynthetic electron transport, is more complex and is regulated by several mechanisms which are activated at different stages of the light stress. Hence, it would be much more convincing to extend the measuring program up to the light intensities causing ETR saturation, ETRmax or PSII photoinhibition, instead of extrapolating these points. Further on, the parameter α (Fig. 5C) and ETRmax (Fig. 5E) and are the components of the same equation hence I do not see much sense in showing them both in separate figures. It would be sufficient to show ETRmax, however with a clear indication that these are “predicted data”. Moreover, α is merely a coefficient which allows for fitting of measured ETR values to the curve therefore I cannot agree for giving it the name “photosynthetic efficiency”. This suggests a more general information about photosynthesis which is untrue. Presentation of calculated values of Ek (panel D), does not bring to any better understanding of differences in PSII quantum efficiency between the two type of needles, therefore I would skip it.

We appreciate the constructive reviewer’s comments on the figure. However, we firmly believe that the current presentation facilitates the understanding of the observed differences between bluish and green trees. In addition, it favors the integration of data from electron transport efficiency with those obtained in physiological and transcriptome analysis.

Description of the 0Y axis in all panels of Fig. 5 should be unified. The names of parameters should be either named in full or abbreviated.

Agreed. The figure has been redrawn to meet the reviewer indications.

Discussion of results in general is too superficial.

We disagree the reviewer's comment if addressed to the full text of the Discussion section. Changes has been made in specific points that we hope to improve the text.

Sentence in l.255-258, with correlation mentioned therein, is not very informative.

Agreed. The sentence has been removed.

One cannot agree with the sentence (l.258-259) that a higher chlorophyll content results in enhanced Fv/Fm. There is not so such straightforward relation between these variables. Moreover, the paper of Sharma et al (2015), cited here, is not present in references.

The referred text does not establish a straightforward relation between these variables, as the reviewers stated.

The paper by Sharma et al (2015) supporting the sentence has been incorporated in the reference list.

The suggestion about changes in thylakoid stacking is too speculative (l.265).

This is a curious comment from the side of the reviewer. Why a suggestion is considered too speculative?

The conclusion given on l. 268 that high ETR values is probably due to a richer pigment composition is not proper.

Agreed. This conclusion has been removed from the text.

The information about loroxanthin in the Discussion (l.276) is confusing. This pigment has been detected so far in algae (van den Berg and Croce 2022: Front Plant Sci, DOI: 10.3389/fpls.2022.797294). Also, the speculation that carotenoids increase the PSII antenna size is misleading and the literature cited there comes only from the study on algae.

The text in the discussion has been rephrased.

The sentence describing the size of PSII antenna and PSI stability (l.282-290) is far too speculative in relation to here obtained results.

The reviewer statement is over demanding. The sentence is just discussing plausible interpretations and suggestions at the light of the obtained results.

The conclusion linking the changes in N metabolism with changes in transcripts of TCA cycle enzymes does also come too far (l.308-309).

This comment is totally unjustified. The linking between N metabolism and TCA cycle enzymes is quite well documented in plants, and particularly in conifers. Please, see Palomo et al. 1998, Plant Physiol, 118: 617-626; Cánovas et al. 2007, J Exp Bot 58: 2307-2318; Foyer et al. 2011, J Exp Bot 62:1467-82.

In regard to the discussion of the data dealing with PSII quantum efficiency I suggest to add more information about the protective function of PSII photoinhibition which might be the case in bluish needles.

Many thanks for the comment. The potential protective function of PSII photoinhibition in bluish trees is already mentioned in the text (lines 288-290).

Discussion of transcriptomic data concerning the chromatin structure and modifications (last paragraph of Discussion) fits better to the part describing changes in transcriptome (at the beginning of the Discussion).

We disagree with the reviewer's suggestion. In our opinion, it is not appropriate to discuss the possible mechanism involved in the phenotypic variation before the molecular and physiological characterization of the phenomenon.

Abstract should be shorten and made more concise.

The abstract has ben modified for a better understanding of the manuscript contents

Reviewer 2 Report

The manuscript presented by Ortigosa and colleagues with the tittle “Transcriptome analysis and intraspecific variation in Spanish 2 fir (Abies pinsapo Boiss.)”.  The main aim of this work is to investigate the intraspecific variations of two populations s of A. pinsapo that can be found within its largest population at the Sierra de las Nieves National Park: one with standard green needles and another with bluish-green needles. To elucidate the causes of both phenotypes, they performed different transcriptome and physiological analyses. They found that the green trees showed higher photosynthetic performance and enhanced levels of chlorophyll, protein and total nitrogen in needles. In sharp contrast, needles of “bluish” trees showed a higher content of carotenoids and cellulose. Then, to investigate the molecular basis of the different responses they performed a transcriptomic analysis of needles of both trees. To do that, they first report a very valuable full transcriptome of Abies pinsapo that was obtained from a variety of plant organs including needles, buds, bark, roots, male and female cones, which thus represent a very interesting resource for future genomic and conservation studies. Then, using the obtained transcriptome as reference, they compare the transcriptome of needles of the two A pinsapo populations and found 457 misregulated genes. Interestingly, they found in green trees, a higher presence of differentially transcripts for proteins involved in substrate transport (transporters) like ammonium transporter, formate/nitrite transporter , allantoin permease,  and transcripts encoding members of the major facilitator superfamily involved in transport of small solutes across cell membranes. Besides, they found transcripts differentially expressed in green trees included those encoding different ribosomal proteins and carbon metabolism suggesting an imbalance in the nitrogen/carbon status of “bluish” trees. Notably, it was also found in green trees, upregulation of key transcripts involved in the chromatin remodeling process such as Swi3, Ada2, N-150 Cor, and TFIIIB suggesting that the observed phenotypic observations could be, at least, partially due to significant epigenomic changes. Finally, the author indicates in the discussion that bluish trees may represent a local adaptation or ecotype to stressing climatic or soil conditions.  In a general comment the results obtained have novelty and great interest and deserve to be published.

In any case, it can be suggested some minor comments:

Results section

Line 90- Indicate reference to table 1 in the main text.

Line 119 Please indicate fold change and p value.

Line 165. Please change the title of the legend of the figure for something more informative like a Functional categorization of the differentially expressed genes in Green vs Bluish trees” or “Gene Ontology analyses…..”!

Line 167. In the foot note of the figure Indicate in the program used for the GO analyses.  

Line 172. It might be included further statistical analyses (correlations) to further support the significant correlation between RNAseq and qRTPCR expression analyses.

Line 181 Figure 3 should include error bars and statistical analyses to demonstrate significant correlation. Please improve the text of the foot note accordingly including the reference genes used in qRTPCR.    

Line 208 Figure 4. The histograms of the figure present different sizes. It could be suggested maintain the same names (Green tree blue tree in x axis) in all figures like in figure 5

Discussion

241-246 Please include a few words to explain transcriptome improvements with previous fir transcriptome available.

324-341. It is mentioned that the bluish phenotype occurs mainly and is more abundant in the lower ecotone of the Abies pinsapo eastern-most populations, in the Yunquera forest. Besides it is also mentioned this area shows one of the 328 driest conditions among the whole A. pinsapo distribution area. It could be useful to the reader indicate if the samples of blue / green trees used in this study are located (map) in specific ecological areas with specific ecological factors like lower water supply, poor soils, altitude, low light?. If there is a positive/negative correlation, then could be interesting to mention and discuss with a few words in the text.

Supp Data set S1. Which are the genes that are not labelled in red or blue in the excel file? Why are those included?

Author Response

Reviewer 2:

Comments and Suggestions for Authors

The manuscript presented by Ortigosa and colleagues with the tittle “Transcriptome analysis and intraspecific variation in Spanish 2 fir (Abies pinsapo Boiss.)”.  The main aim of this work is to investigate the intraspecific variations of two populations s of A. pinsapo that can be found within its largest population at the Sierra de las Nieves National Park: one with standard green needles and another with bluish-green needles. To elucidate the causes of both phenotypes, they performed different transcriptome and physiological analyses. They found that the green trees showed higher photosynthetic performance and enhanced levels of chlorophyll, protein and total nitrogen in needles. In sharp contrast, needles of “bluish” trees showed a higher content of carotenoids and cellulose. Then, to investigate the molecular basis of the different responses they performed a transcriptomic analysis of needles of both trees. To do that, they first report a very valuable full transcriptome of Abies pinsapo that was obtained from a variety of plant organs including needles, buds, bark, roots, male and female cones, which thus represent a very interesting resource for future genomic and conservation studies. Then, using the obtained transcriptome as reference, they compare the transcriptome of needles of the two A pinsapo populations and found 457 misregulated genes. Interestingly, they found in green trees, a higher presence of differentially transcripts for proteins involved in substrate transport (transporters) like ammonium transporter, formate/nitrite transporter , allantoin permease,  and transcripts encoding members of the major facilitator superfamily involved in transport of small solutes across cell membranes. Besides, they found transcripts differentially expressed in green trees included those encoding different ribosomal proteins and carbon metabolism suggesting an imbalance in the nitrogen/carbon status of “bluish” trees. Notably, it was also found in green trees, upregulation of key transcripts involved in the chromatin remodeling process such as Swi3, Ada2, N-150 Cor, and TFIIIB suggesting that the observed phenotypic observations could be, at least, partially due to significant epigenomic changes. Finally, the author indicates in the discussion that bluish trees may represent a local adaptation or ecotype to stressing climatic or soil conditions.  In a general comment the results obtained have novelty and great interest and deserve to be published.

Thank you for the critical evaluation of our work

In any case, it can be suggested some minor comments:

Results section

Line 90- Indicate reference to table 1 in the main text.

Done

Line 119 Please indicate fold change and p value.

This requested information is provided in the section of Materials and Methods:

“Only the transcripts with |logFC| > 1 and FDR < 0.05 were considered as differentially expressed (DE).”

Line 165. Please change the title of the legend of the figure for something more informative like a Functional categorization of the differentially expressed genes in Green vs Bluish trees” or “Gene Ontology analyses…..”!

Many thanks for the comment.

Agreed.  The title of the legend has been changed  for something more informative

Line 167. In the foot note of the figure Indicate in the program used for the GO analyses.

The program used for the GO analyses is now indicated in the foot note of the figure.

Line 172. It might be included further statistical analyses (correlations) to further support the significant correlation between RNAseq and qRTPCR expression analyses.

Agreed. The figure has been modified to include the correlation between RNAseq and qRTPCR expression analyses.

Line 181 Figure 3 should include error bars and statistical analyses to demonstrate significant correlation. Please improve the text of the foot note accordingly including the reference genes used in qRTPCR.

This problem has been fixed

Line 208 Figure 4. The histograms of the figure present different sizes. It could be suggested maintain the same names (Green tree blue tree in x axis) in all figures like in figure 5

Done

Discussion

241-246 Please include a few words to explain transcriptome improvements with previous fir transcriptome available.

The previous work by Pérez-González et al. (2018) Forest Science 64, 6, reports the Illumina transcriptome sequencing of a limited number of samples from pinsapo seedlings. Therefore, “sensu stricto” the sequence information available cannot be considered representative of a reference transcriptome of the tree. In addition, the cover of full transcripts is quite limited and insufficient to carry out functional and conservation studies.  We prefer do not assess this previous work in the text of our paper,  the magnitude of the data provided here is self-explanatory.

324-341. It is mentioned that the bluish phenotype occurs mainly and is more abundant in the lower ecotone of the Abies pinsapo eastern-most populations, in the Yunquera forest. Besides it is also mentioned this area shows one of the 328 driest conditions among the whole A. pinsapo distribution area. It could be useful to the reader indicate if the samples of blue / green trees used in this study are located (map) in specific ecological areas with specific ecological factors like lower water supply, poor soils, altitude, low light?. If there is a positive/negative correlation, then could be interesting to mention and discuss with a few words in the text.

This is a very interesting comment. Unfortunately, no additional information is currently available. However, we hope that further studies can be performed in the coming years on the ecological characteristics of specific areas in which the two variants of pinsapo fir coexist.

Supp Data set S1. Which are the genes that are not labelled in red or blue in the excel file? Why are those included?

We decided to include the complete list of differentially expressed genes.
